# Generalized Linear Models to Forecast Malaria Incidence in Three Endemic Regions of Senegal

**DOI:** 10.3390/ijerph20136303

**Published:** 2023-07-05

**Authors:** Ousmane Diao, P.-A. Absil, Mouhamadou Diallo

**Affiliations:** 1ICTEAM Institute, UCLouvain, B-1348 Louvain-la-Neuve, Belgium; pa.absil@uclouvain.be; 2Molecular Biology Unit/Bacteriology-Virology Lab, CNHU A. Le Dantec/Université Cheikh Anta Diop, Dakar Fann P.O. Box 5005, Senegal; mouhamdiallo@gmail.com

**Keywords:** epidemiological data, meteorological data, generalized linear models, parameters estimation, forecasting

## Abstract

Affecting millions of individuals yearly, malaria is one of the most dangerous and deadly tropical diseases. It is a major global public health problem, with an alarming spread of parasite transmitted by mosquito (Anophele). Various studies have emerged that construct a mathematical and statistical model for malaria incidence forecasting. In this study, we formulate a generalized linear model based on Poisson and negative binomial regression models for forecasting malaria incidence, taking into account climatic variables (such as the monthly rainfall, average temperature, relative humidity), other predictor variables (the insecticide-treated bed-nets (ITNs) distribution and Artemisinin-based combination therapy (ACT)) and the history of malaria incidence in Dakar, Fatick and Kedougou, three different endemic regions of Senegal. A forecasting algorithm is developed by taking the meteorological explanatory variable $X_{j}$ at time $t-{𝓁}_{j}$, where *t* is the observation time and ${𝓁}_{j}$ is the lag in $X_{j}$ that maximizes its correlation with the malaria incidence. We saturated the rainfall in order to reduce over-forecasting. The results of this study show that the Poisson regression model is more adequate than the negative binomial regression model to forecast accurately the malaria incidence taking into account some explanatory variables. The application of the saturation where the over-forecasting was observed noticeably increases the quality of the forecasts.

## 1. Introduction

Malaria is a disease caused by a parasitic infection transmitted by a mosquito (female Anophele). It can also be passed to human by blood transfusion, sharing needles or congenitally [1]. According to the 2020 World Health Organization (WHO) report, malaria caused 627,000 deaths, 95% of which were registered in African Region. The number of malaria cases decreased from 2000 (238 million reported cases) to 2019 (229 million reported cases). In spite of this diminution, malaria is still endemic in many countries in the world, particularly in Africa. In Senegal, it constitutes a major public health problem, according to the “Programme national de lutte contre le paludisme (PNLP)”.

Several mathematical or statistical models were developed for predicting malaria case incidence. Generalized Linear Models (GLM) [2,3,4] were used in the literature. Examples of GLM include the Poisson regression developed first by Nelder and Wedderburn [5], the negative binomial (NB) regression [3], the quasi-Poisson regression [5] and the zero-inflated regression [6]. In general, the Poisson regression is very popular for data fitting but its mean-equal-variance property can limit its application on over-dispersed data [5,6,7]. Also in [8], a multivariate generalized Poisson regression model was defined and studied.

In [6], a model that adapts to malaria incidence using the zero-negative binomial was developed based on climate variables and mosquito density in Limpopo province, South Africa. The results in [6] show how rain and average temperature affect the incidence of malaria. In [9], authors introduced a model that takes into account the incidence of malaria morbidity and mortality in Akure, Nigeria. In that work, the negative binomial regression model, with log as link function, was used to express the malaria morbidity and mortality incidence as functions of climatic variables. Then, the autoregressive integrated moving average (ARIMA (*p*, *d*, *q*)) model was used to fit the residuals. The findings in [9] revealed that an increase in minimum temperature and relative humidity at a 1-month lag significantly increases the chance of malaria transmission and thereby leads to an increase in the number of inpatient and outpatient individuals, as well as the total number of malaria cases. In another study [10], a Bayesian spatiotemporal analysis has been made to describe year-to-year variation of malaria incidence data from Zimbabwe, and in relation to variation in climate risk factors to enhance our ability of developing an operational malaria early warning system (MEWS) and determine areas prone to climate-driven epidemics. As methods in [10], the authors used the annual proportion of monthly malaria cases and Markham’s seasonality index to display between-year variation in the data. Then, the data were fitted with the Bayesian negative binomial models such as the non-spatial model, the spatial model and the spatiotemporal model. In addition, a Markov Chain Monte Carlo (MCMC) simulation was applied to estimate the model parameters. As a result in [10], it was found that a high annual malaria incidence coincides with high rainfall and relatively warm conditions while low incidence years coincide only with low rainfall. In conclusion, all models indicated that the mean annual temperature, rainfall, vapour pressure and normalized difference vegetation index (NDVI) were strong positive predictors of increased annual incidence rate. In [11], the authors applied and compared a Bayesian and classical methods of parameter estimation on the effect of climatic factors in the context of modelling malaria incidence in Limpopo Province, South Africa. In that work, the authors estimated the parameters from a negative binomial model by a Bayesian estimation and maximum likelihood estimation. As result, in [11] the Bayesian method appears more robust than the classical method in analyzing malaria incidence. In [12], the authors include the link between CD4 cell count and influencing covariates of biometric and demographic factors from negative binomial mixed models. A GLM is applied in [13] where authors provide spatially explicit burden estimates of malaria in Senegal using the Senegal Malaria Indicator Survey (SMIS) data and Bayesian geostatistical Zero-Inflated Binomial models based on variable selection methods for spatial data.

A comparative study of existing models was carried out across six countries of Sub-Saharan Africa—Burkina Faso, Nigeria, DRC, Mali, Cameroon, and Niger—over a period of 28 years on malaria incidence in [14]. It is reported in [14] that the SARIMA model was found to work best with time series data that exhibited periodic or seasonal characteristics and was able to predict the seasonal trend of malaria. That model type is only suitable for a stationary or seasonal process. The negative binomial model correctly identified the association between climate variables (taken as explanatory variables) and the rate of malaria transmission. That last model type can make good short-term forecasts, but is not ideal for prediction in subsequent years.

In this paper, a GLM is used in the context of forecasting falciparum malaria incidence count per month based on climate variables and history of falciparum malaria incidence count per month in three endemic regions of Senegal: Dakar (hypoendemic zone), Fatick (endemic zone), and Kedougou (hyperendemic zone). The choice of these three regions is motivated by data availability (notably the presence of villages where longitudinal studies have been conducted), but also by geographical differences: influence of the ocean in the Dakar peninsula, tropical climate in Kedougou, and savanna landscape in Fatik. These fundamental geographical differences allow us to test the applicability of GLMs under drastically different evolutions of the climate variables. A machine learning approach is developed, based on a separation of the data into a train-set to estimate the parameters by maximum likelihood and a test-set to assess the forecast accuracy. Addition and ablation studies are developed to show the influence of each explanatory variable in the forecasts. A Vuong test reveals that the Poisson distribution is preferred to the negative binomial distribution to model the malaria incidence given the explanatory variables. The forecast accuracy of GLMs with various distributions (Poisson, negative binomial, and Gaussian) and link functions (identity, log, and sqrt) is compared in terms of several model performance metrics. Whereas the best distribution-link combination varies according to the endemic region of interest and the performance metric, the experiments lead to the conclusion that the Poisson distribution with the identity link is overall the most suitable combination. In addition, a saturation method is introduced on the rainfall variable to remedy some overestimations observed during the forecasts. This method has reduced by 4% in the sense of MARE, the over-estimation occurring at the end of 2015 in Dakar.

The paper is organized as follows. The available data are presented in Section 2.1. The models are described in Section 2.2. The estimation and forecasting method containing the train-test machine learning method, the principles of forecasting, the algorithmic protocols, and the saturation concept are presented in Section 2.3. Experimental results and discussion are reported in Section 3 and conclusions are drawn at the end.

## 2. Materials and Methods

### 2.1. Data and Notation

There are two principal malaria transmission zones in Senegal:“Faciès tropical”: corresponding the regions of Ziguinchor, Kolda, Tambacounda, and Kedougou. In that zone, the raining season is the longest and most intensive in the country and covers 5 to 6 months. Malaria cases are observed between 4 to 6 months and the transmission is high (20 to 100 infected bites/human/year).“Faciès sahélien”: corresponding the regions such as Kaolack, Fatick, Diourbel, Dakar, Thies, Louga, Saint-Louis, and Matam with a less intensive rainy season and covers 2 to 3 months. The transmission is very low in general (0 to 20 infected bites/human/year).

We are interested in three regions of Senegal: Dakar (hypoendemic zone), Fatick (endemic zone), and Kedougou (hyperendemic zone) located in the map presented in Figure 1.

The historical data, such as the monthly falciparum malaria incidence count, the distributed insecticide-treated bed-nets (ITNs), and the distributed Artemisinin-based combination therapy (ACT), between 2008 and 2016, come from the “Programme national de lutte contre le paludisme (PNLP)” of Senegal (https://www.dropbox.com/s/0p4uc2dihfhr9cb/Dakar.csv?dl=0 (accessed on 3 July 2023)). In this study, we consider as malaria cases, the cumulative number of confirmed tests by Rapid diagnostic tests (RDTs) during the month, in all individual groups [15,16]. Malaria cases are confirmed by the methods validated by the “Programme national de lutte contre le paludisme (PNLP)” of Senegal in accordance to the WHO guidelines. The main method is the rapid diagnostic test (RDT) even if it has been recently discovered that this test could miss up to 20% of malaria cases [17]. Due to unavailability of the RDT in some deep localities and the lack of materials to keep it, some districts use the “goutte épaisse”, which is a very old method and less sensitive than the RDT. There also is a more sensitive but very expensive test: polymerase chain reaction (PCR), used only in some high level medical research institute in Senegal. Then, we grouped the data from all the different big sanitary districts in each region such as Dakar, Fatick, and Kedougou. The hourly meteorological data, such as the temperature, the relative humidity, and the rainfall, between 2008 and 2016, come from meteoblue (https://www.meteoblue.com/historyplus (accessed on 3 July 2023)). To obtain adequate meteorological data for our study (monthly time unit) we add up all the measured values of the month for rainfall, and we calculate the mean of all the measured values of the month for temperature and humidity.

#### 2.1.1. Response Variable

In this study, the response variable (or explained variable) is the falciparum malaria incidence count per month noted by $Y_{o}\left(t\right)$.

#### 2.1.2. Independent Variables

The available explanatory variables of this study are the history of falciparum malaria incidence count per month ($Y_{o}\left(t-\delta \right)$), the rainfall (R(t−δ), mm per month), the average temperature (T(t−δ), °C per month), the relative humidity (H(t−δ), % per month), the number of insecticide treated bed-nets distributed per month (B(t−δ)), the number of anti-malarial drugs distributed per month (A(t−δ)) where we consider δ=h,h+1,…,6 (*h* represents a forecast horizon), and an artificial vector that we call intercept vector *I* equal to 1 all *t*.

We are interested in the meteorological explanatory variables such as the rainfall (*R*), the average temperature (*T*) and the relative humidity (*H*) because they are known to influence the mosquitoes (Anopheles) ecology by affecting its distribution, seasonality, and transmission intensity [18,19]. For example, the temperature influences the sporogonic development duration of the parasite and many parameters related to the mosquitoes such as: the biting rate, the egg deposition rate, and the death rate of immature and adult mosquitoes [19]. The rainfall influences the availability and the quality of the larval breeding grounds [20] and the maturation of immature mosquitoes [19]. As for the bed-net (*B*) and the drugs (*A*) distributed, we took them because we suppose that they constitute the main factors fighting against malaria [16], reducing the morbidity and the mortality of the disease. Needless to say, the number of bed-nets actually used would be a more suitable explanatory variable, but these data are not available. Note that, in contrast with the physical explanatory variables (*R*, *T*, and *H*), the human explanatory variables (*B* and *A*) may depend on the incidence in the past, and they may also be influenced by models used by the health authorities. For this reason, in most of our experiments, we only consider the physical explanatory variables.

All simulations and data analysis are carried out with Jupyter Notebook (Anaconda 3) and Spyder (Python 3.7). Figure 2, Figure 3, Figure 4, Figure 5, Figure 6 and Figure 7 (reproduce with Malaria_inci_and_variable_2022_07_23.py) present the plots of some variables in order to illustrate their annual distribution. Figure 2, Figure 3 and Figure 4 show the variations of malaria cases in relation to the rainfall and Figure 5, Figure 6 and Figure 7 show the variations of malaria cases in relation to the bed-net.

Our experiments use the epidemiological data from PNLP and the meteorological data from meteoblue from 31 January 2008 to 31 December 2016, in Dakar, Fatick and Kedougou.

### 2.2. Models

We have *n* data points $\left(X_{t1},X_{t2},...,X_{tk},Y_{t}\right)\in R^{k+1}$ for t=1,...,n where *k* is the number of explanatory variables (including the intercept vector) and *n* is the number of months. We want to build a generalized linear model (GLM) of the response vector *Y* using the *k* explanatory variables $X_{1},...,X_{k}$, according to the diagram Equation (Equation 1), where we denote by R-v: Random variable, L-f: Link function and D-c: Deterministic component. According to [3], the link function permits the mean (μ) of the *t* th observation and its linear predictor (η) to be related. We let $X_{t1}=1$, t=1,…,n making $\beta_{1}$ the intercept. In the D-c block, the regression coefficients $\beta_{1},\ldots ,\beta_{k}$ are to be estimated on the train-set.
(1)$$\mathrm{R-v}:Y_{t}∼f\left(Y_{t};\mu_{t}\right)\overset{\mu_{t}}{⟵}\mathrm{L-f}:g\left(\mu_{t}\right)=\eta \left(X_{t}\right)\overset{\eta }{⟵}\mathrm{D-c}:\eta \left(X_{t}\right)=\sum_{j=1}^{k}\beta_{j}X_{tj}.$$

According to the studies in [3,12,21], candidate distributions for viable modeling include the Poisson and negative binomial (NB) distributions. Now, we are going to present these two regression models in the following Section 2.2.1 and Section 2.2.2.

#### 2.2.1. Poisson Regression Model

The Poisson distribution is probably the most used discrete distribution because of its simplicity, according to [11]. Its conditional probability mass function is defined as in [2,11] by
$$\begin{array}{ccc} f(Y_{t};\mu_{t}) & = & \frac{\mu_{t}^{Y_{t}}\exp\left(-\mu_{t}\right)}{Y_{t}!}\\ & = & \exp[Y_{t}\log\left(\mu_{t}\right)-\mu_{t}-\log\left(Y_{t}!\right)]. \end{array}$$

According to Equation (Equation 1), we have
(2)$$\begin{array}{ccc} \mu_{t} & = & g^{-1}\left(X_{t}^{T}\beta \right)\\ \; & = & g^{-1}\left(\beta_{1}X_{t1}+\ldots +\beta_{k}X_{tk}\right). \end{array}$$

#### 2.2.2. NB Regression Model

The Poisson–Gamma mixture distribution is the negative binomial distribution, according to [6]. Its probability mass function is given as in [5,6,7] by
(3)$$f\left(Y_{t};\mu_{t},\alpha \right)=\frac{\Gamma (Y_{t}+\frac{1}{\alpha })}{\Gamma \left(\frac{1}{\alpha }\right)\Gamma \left(Y_{t}+1\right)}{\left(\frac{1}{1+\alpha \mu_{t}}\right)}^{\frac{1}{\alpha }}{\left(\frac{\alpha \mu_{t}}{1+\alpha \mu_{t}}\right)}^{Y_{t}},$$
where Γ is the gamma function. Its mean is $\mu_{t}$ and its variance is $\mu_{t}+\alpha \mu_{t}^{2}$, where α is termed the distribution parameter. Note that, if α→0, the negative binomial converges to the Poisson distribution.

The Section Appendix A describes how the regression coefficients are computed.

### 2.3. Estimation and Forecasting Methods

#### 2.3.1. Train and Test Sets

We have $t_{s}<t_{i}<t_{c}<t_{e}$, where $t_{s}$ is the initial time of the data, $t_{i}$ is the initial time of the observed malaria incidence, $t_{c}$ is the end time of the train set, $t_{e}$ is the end time of the test set.

#### 2.3.2. Parameter Estimation and Principles of Forecasting

We train the model by taking the observed malaria incidence ($Y_{o}\left(t\right)$) in $\left[t_{i},t_{c}\right]$ and the explanatory variables ($X_{tj},j=1,\ldots ,k$) in $\left[t_{i}-\delta ,t_{c}-\delta \right]$. We did this in order to have the regression coefficients βs and the dispersion parameter α (only with NB regression model).

In [2], the authors suggest to calculate α using a technique that they call auxiliary Ordinary Least Squares (OLS) regression without a constant. In the negative binomial case, a first estimation of the βs is obtained by the procedure of the Poisson case. Then, α is computed by OLS method. Finally, the βs are re-estimated by maximizing the log likelihood (Equation (Equation 20)) wrt β.

Then, we make the forecasts, according to Algorithm 1, with the coefficients found in the train period. We assess the model accuracy in the test period $\left[t_{c}+1,t_{e}\right]$ by comparing the theoretical (mean $\mu_{t}$) and the observed ($Y_{o}\left(t\right)$) incidences.

Algorithm 1 describes the train-test procedure. For the link function *g*, the choices identity, log and sqrt are available in the Python library statsmodels.genmod.families.links. The forecasts are obtained with the formula
(4)$$\mu_{t}=g^{-1}\left(\beta_{1}X_{1}\left(t\right)+\sum_{j=2}^{k}\beta_{j}X_{j}\left(t-\delta \right)\right).$$

**Algorithm 1:** Forecasting Algorithm
**Input:** $t_{s}$, $t_{i}$, $t_{c}$, $t_{e}$ ← times of Section 2.3.1;
              *ℓ* ← vector (Section 3.1);
              *h* ← forecast horizon (h≥1);
              $Y_{o}$ ← observed malaria incidence (dependant variable);
              $X=\left\{X_{1},X_{2},\ldots ,X_{k}\right\}$ ← set of explanatory variables;
              f←distribution(PoissonorNB);
              g←link(identity,log,orsqrt);
**Output:** $\hat{Y}$ ← the forecasted vector of malaria incidence;


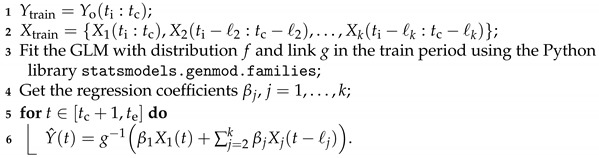




#### 2.3.3. Saturation Method

We would like to test a saturation in the explanatory variables, in particular rainfall. The motivation is that additional rainfall should have less impact on the malaria incidence in a wet period than in a dry period. The saturation can be simply a hyperbolic tangent function. We posit that, instead of being linear, the contribution of rainfall to η(X) is affected by a saturation, which we model by
(5)$$R_{\mathrm{sat}}=\gamma \tanh\left(R/\gamma \right).$$

We estimate the parameter γ by making a research in many initial values of γ in order to find the more adapted value which gives the low RMSE_train after fitting the GLM. This procedure is entirely described in Algorithm 2.
**Algorithm 2:** Forecasting Algorithm with Saturation**Input:** $t_{s}$, $t_{i}$, $t_{c}$, $t_{e}$ ← times of Section 2.3.1;              *ℓ* ← vector (Section 3.1);              *h* ← forecast horizon (h≥1);              $Y_{o}$ ← observed malaria incidence (dependant variable);              $X=\left\{X_{1},X_{2},\ldots ,X_{k-1},R\right\}$ ← set of explanatory variables;              f←distribution(PoissonorNB);              g←link(identity,log,orsqrt);**Output:** $\hat{Y}$ ← the forecasted vector of malaria incidence;
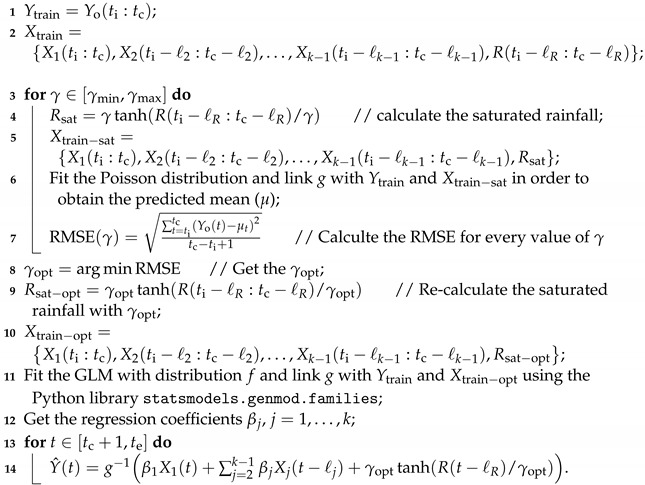


## 3. Results and Discussion

In this section, we first present and discuss the results of the correlation between the explained variable and each explanatory variable. We define metrics. Then, we present the forecast results from the Algorithm 1. Finally, we present the results from other methods such as addition study, ablation study, and saturation.

### 3.1. Determination of Lags

For Dakar data in Figure 2, we observe, every year, an increase in malaria cases in the rainy season (from May or June to October or November), reaching a peak around the month of October or November, and a decrease to become stationary along the dry season. This situation is also observed in Fatick (Figure 3) and Kedougou (Figure 4), and proves the seasonality of the malaria cases from these three regions.

The plots of the malaria cases and the explanatory variables (e.g., Figure 2, Figure 3 and Figure 4) reveal that there is a delay between the maximum of malaria cases and the maximum of the explanatory variable) in the three data sets. This delay is called lag and represented by *ℓ* in the formulae. We set
(6)$${𝓁}_{j}:=\underset{\delta \in \{h,h+1,\ldots ,6\}}{\arg\max}\left|r(Y_{o}\left(t_{i}:t_{e}\right),X_{j}\left(t_{i}-\delta :t_{e}-\delta \right))\right|,$$
where *r* denotes the sample Pearson correlation coefficient. The statistical results are presented in Table 1 (reproduce with Determination_of_lag_2022_07_23.py) and the evolution of correlations as function of lags is illustrated in Figure 8, Figure 9 and Figure 10 (reproduce with Correlation_Plots_2022_07_23.py).

Since these *p*-values are less than 0.05, we could conclude that there is a statistically significant correlation between the malaria incidence and the explanatory variable at the delay considered. The only exception is the bed-nets distributed in Dakar where the *p*-value is 0.129 > 0.05.

The rainfall is highly and positively correlated, meaning that if the rainfall increases, the malaria incidence will increase. These results are also found in [11], revealing that rainfall was a strong positive predictor of increasing the annual incidence rate. The bed-net distribution is weakly correlated in Dakar. Then, it is positively correlated in Fatick (even if the value is low) and in Kedougou with a lag of three months. Indeed, the situation in Fatick and Kedougou can be explained by the fact that the authorities anticipate the bed-net distribution three months before the beginning of the rainy season. On the other hand, in Figure 7, we observe that the bed-net distribution is not regular in Kedougou because its behavior is very different in the last two years of the dataset. For all these reasons, we do not use the bed-net distribution (*B*) as an explanatory variable. The optimal lag between the malaria cases and the drugs (*A*) is equal to 0 in all the regions. This result means that the drugs is not a real predictor because it is usually taken after appearing some malaria symptoms from a patient. That reason leads us to do not consider the drug as an explanatory variable. But it can help to cure some sick people and to reduce future malaria cases. In Figure 8, Figure 9 and Figure 10, the correlation between malaria cases at time *t* and the malaria in the past (t−δ) decreases for all values of δ meaning that the malaria in the past, as an explanatory variable, becomes less and less important when δ becomes larger. The explanatory variables for Y(t) are thus finally $Y_{o}\left(t-{𝓁}_{Y_{o}}\right),R\left(t-{𝓁}_{R}\right),T\left(t-{𝓁}_{T}\right),H\left(t-{𝓁}_{H}\right)$, and 1 for the intercept.

### 3.2. Model Performance Metrics

The output of the GLM is a probability distribution at each time instant. Accuracy measures such as the root mean square error (RMSE) and the mean absolute scaled error (MASE) defined in [22], and the mean absolute relative error (MARE) defined in [23], quantify the discrepancy between the mean $\mu_{t}$ of the distribution and the observed incidence $Y_{o}\left(t\right)$. We also introduce other statistical measures such as the scatter index (SI) and the reliability analysis (RA) defined in [24]. We conduct experiments for 108 months with a train and a test period.

Root mean square error (RMSE):
(7)$$\mathrm{RMSE}=\sqrt{\frac{\sum_{t=t_{i}}^{t_{c}}{\left(Y_{o}\left(t\right)-Y\left(t\right)\right)}^{2}}{t_{c}-t_{i}+1}}.$$Mean absolute error (MAE):
(8)$$\mathrm{MAE}=\frac{1}{t_{c}-t_{i}+1}\sum_{t=t_{i}}^{t_{c}}\left|Y_{o}\left(t\right)-Y\left(t\right)\right|.$$Mean absolute scaled error (MASE):
(9)$$\mathrm{MASE}=\frac{\mathrm{MAE}}{\frac{1}{t_{c}-t_{i}}\sum_{t=t_{i}}^{t_{c}-1}\left|Y_{o}\left(t+1\right)-Y_{o}\left(t\right)\right|}.$$It consists of the ratio between the MAE and the mean monthly variation of the observed values. A MASE value around 1 or below indicates an excellent accuracy.Mean absolute relative error (MARE):
(10)$$\mathrm{MARE}=\frac{1}{t_{c}-t_{i}+1}\sum_{t=t_{i}}^{t_{c}}\frac{|Y_{o}\left(t\right)-Y\left(t\right)|}{|Y_{o}\left(t\right)|}.$$R-squared [2]:
(11)$$R_{\mathrm{COR}}^{2}={\left(C\hat{O}R\left[Y_{o},\mu \right]\right)}^{2}.$$It is the proportion of variation in the outcome that is explained by the predictor variables. The higher the R-squared, the better the model, in contrast to all the above metrics.

The Scatter index (SI) (also called the normalized root mean squared error (NRMSE)) and the reliability analysis (RA) are defined in [24].

The SI presents the percentage of RMSE difference with respect to mean observation or it gives the percentage of expected error for the parameter. Lower values of the SI are an indication of better model performance.
(12)$$\mathrm{SI}=\frac{\sqrt{\left(1/n\right)\sum_{t=t_{a}}^{t_{b}}{\left(\left(Y_{t}-\bar{Y}\right)-\left(Y_{o}\left(t\right)-\bar{Y_{o}}\right)\right)}^{2}}}{\sqrt{\left(1/n\right)\sum_{t=t_{a}}^{t_{b}}Y_{o}\left(t\right)}}.$$The reliability analysis (RA) is a statistical method for measuring the overall consistency of a model by determining if this suggested model achieves a permissible level of performance.
(13)$$\mathrm{RA}=\left(\frac{100\%}{t_{b}-t_{a}+1}\right)\sum_{t=t_{a}}^{t_{b}}k\left(t\right),$$
where the *k*s are determined through two steps. First, the relative average error (RAE) is defined as a vector whose *t*th component is
(14)$$\mathrm{RAE}\left(t\right)=|\frac{Y_{o}\left(t\right)-Y\left(t\right)}{Y_{o}\left(t\right)}|.$$Next, if RAE(t)≤Δ, then k(t)=1, otherwise k(t)=0, where Δ is a threshold value that is 0.2 (20%) based on Chinese standards.

### 3.3. Model Selection and Result Comparison

#### 3.3.1. Model Selection by Using the Vuong Test

In order to assess the adequacy of the distribution, we apply the Vuong statistical test as in [25,26]. The Vuong test is defined as follows:(15)$$V=\frac{\sqrt{n}\frac{1}{n}\sum_{t}^{n}m_{t}}{\sqrt{\frac{1}{n}\sum_{i}^{n}{\left(m_{t}-\bar{m}\right)}^{2}}},$$
where $m_{t}=\log\left(f_{1}\left(Y_{t}|X_{t}\right)/f_{2}\left(Y_{t}|X_{t}\right)\right)$ in which $f_{1}\left(Y_{t}|X_{t}\right)$ is the first probability mass function and $f_{2}\left(Y_{t}|X_{t}\right)$ is the second probability mass function. If V>1.96, then the first model is preferred. If V<−1.96, then the second one is preferred. If −1.96<V<1.96, none of the models are preferred. In our case, we let $f_{1}$ be the Poisson distribution and $f_{2}$ the NB distribution. This statistical test permits to choose the most adequate between the two regression models in order to fit the data.

Figure 11, Figure 12 and Figure 13 (reproduce with Vuong_test_2022_07_23.py) present the dependence of the Vuong test value (*V*) with respect to α (dispersion parameter). The figures show that the α computed by OLS is usually in the window where the Poisson model is preferable to the NB model. An exception is observed in Kedougou where none is preferred with log and sqrt. We can conclude that the use of the GLM with Poisson distribution is justified.

#### 3.3.2. Results Comparison by Using Metrics

In this part, we made a comparative analysis in Table 2 (reproduce with Comparative_results_with_all_models_2022_07_23.py) between the three models that are Gaussian (identity, log), Poisson (identity, log, sqrt), and negative binomial (identity, log, sqrt). Experiments are carried out with Algorithm 1 without saturation. We include the minimum values of the predicted mean obtained after fitting the model. A negative sign indicates that the model is not adequate for our count data. All these metric values permit us to validate and to compare the performance of the models.

Results in Table 2 show that the most of the MASEs and MAREs in the train and test windows are around 1 in the three regions. Then, we also have the $R_{\mathrm{COR}}^{2}$s indicator whose values are high (>50%) indicating a good contribution of the explanatory variables in the forecasts. Additionally, predictions provided by Poisson (identity) are globally more reliable in in the three regions when compared to other models based on RA values.

We can conclude that the Poisson (identity) model can be nicely used to fit our data for parameter estimation and to make the forecasts with these found parameters in the three regions. All the additional studies will base on this model.

### 3.4. Forecasts Results by Various Sets of Explanatory Variables

Using Algorithm 1 with Poisson for *f*, identity for *g*, $t_{s}=0$, $t_{i}=5$, $t_{c}=84$, $t_{e}=108$, and h=1, we validate the model with datasets from Dakar, Fatick, and Kedougou.

#### 3.4.1. Forecasts Results Using History of Malaria Incidence Only

For transmissible diseases, the incidence at a given time t−δ is very important to predict the expected incidence at a given time *t*. That reason leads us to first consider a set of explanatory variables composed of two variables that are the history of malaria incidence ($Y_{o}\left(t-\delta \right)$) and the intercept vector (I(t)). We defined this set as follows:(16)$$\mathrm{start}\text{\_}\mathrm{set}\leftarrow \{Y_{o}\left(t-\delta \right),I\left(t\right)\}.$$

Thus, our linear predictor becomes like a simple Markov model with the Gaussian distribution that is called first order autoregressive AR(1) [3] and defined by
(17)$$\mu_{t|t-\delta }=g^{-1}\left(\beta_{1}I\left(t\right)+\beta_{2}Y_{o}\left(t-\delta \right)\right).$$

The estimates returned by the model in Figure 14, Figure 15 and Figure 16 are very accurate because of the low standard errors and the tight 95% confidence interval and statistically significant due to the *p*-values that are less than 0.05. That is why we decide not to show the confidence intervals of the predictions as they do not clearly appear. In Figure 14, Figure 15 and Figure 16, (reproduce with Addition_study_2022_07_23.py) we present the forecast results (noted by A) and the curves of $\beta_{j}X_{j}$ (noted by B). These results are obtained from Equation (Equation 17) with Poisson (identity) model when we take h=1. The figures noted by B permit to show which variable is highly ($\beta_{j}X_{j}\left(t\right)\gg 0$) or weakly ($\beta_{j}X_{j}\left(t\right)∼0$) used during the forecasting process. In all three regions, we observe that the sole use of the history of malaria incidence at time t−1 gives good results. That can be explained by the fact that it is highly used based on figures (B) compared to the intercept whose values are close to 0. This situation is biologically true because with the transmissible diseases like malaria the history of cases is very important to estimate the future cases.

In addition, in Table 3, (reproduce with Various_forecasting_horizon_2022_07_23.py) we remark that the model gives less accurate predictions when the values of *h* increases corresponding a weak use of the history of malaria incidence ($Y_{o}\left(t-h\right),h>1$). We also have collected, in Table 3, the result of SI in the training and testing periods with various value of *h*. These values of SI reveal that the model is the most accurate when h=1 because they are the lowest in all the three regions.

The conclusion of these results is that malaria in the past was a very good explanatory variable when h=1.

#### 3.4.2. Forecasts Results by Using all Explanatory Variables

In this part, we use the whole explanatory variables and the results from Algorithm 1 and Equation (Equation 4) are presented in Figure 17, Figure 18 and Figure 19 (reproduce with Saturated_and_non_ saturated_rainfall_2022_07_23.py). The confidence intervals of the predictions are not shown based on the same observations made in the previous Section 3.4.1.

Figure 17B of Dakar data shows with some peaks in the rainfall distribution. While, in Figure 18B about Fatick data, the rainfall variable is weakly used and this variable is negatively used in Figure 19 of Kedougou data. The peaks observed in Dakar and the situation in Kedougou leads us to develop the method of saturation in Section 2.3.3. Then, in Figure 17B of Dakar data, it is also shown that the humidity is weakly used to mean that this variable has a little contribution in the forecasts. Contrary to Figure 18B about Fatick data, the humidity is highly used, so it has a big participation in the forecasts. As for Kedougou, in Figure 19B, this variable (humidity) is moderately used. As for temperature, its contribution changes depending on the region. For example, in Dakar and Fatick, this variable is negatively used as shown in Figure 17 and Figure 18. However, in Kedougou, it is positive and highly used in Figure 19.

We can summarize here that each explanatory variable does is not used the same in the three regions. Thus, these analyses and observations lead to the development of the methods in Section 3.5 and Section 3.6.

### 3.5. Addition Study

In this section, we experiment with different combinations of sets from Equation (Equation 16) and we define them as follows
$$\begin{array}{c} \{\mathrm{start}\text{\_}\mathrm{set}\cup X_{j},j=3,\ldots ,k\}. \end{array}$$

The purpose is to investigate the influence of each additional variable compared to the observation in Section 3.4.1.

The Table 4 (reproduce with Addition_study_2022_07_23.py) reveals that, in Kedougou, forecast results are improved by adding to start_set explanatory variables such as temperature or humidity. In contrast, the addition of the rainfall (*R*) gives some less good results in the sense of the RMSE, MASE and MARE, in Dakar and Fatick, even if the higher values of $R_{\mathrm{COR}}^{2}$ are observed there.

### 3.6. Ablation Study

We now present ablation studies where we start with all the available explanatory variables and investigate the effect of removing any one of them. The sets of explanatory variables after ablation are thus
$$\begin{array}{c} \left\{X_{1},X_{2},\ldots ,X_{k}\right\}\setminus X_{j},j=2,\ldots ,k. \end{array}$$

By doing that, we can know which variable is more responsible of the over-forecasting (or under-forecasting) observed.

The results in Table 5 (reproduce with Ablation_study_2022_07_23.py) show a low accuracy in terms of RMSE, MASE, and MARE in the whole three regions data sets when the malaria incidence in the past was deleted. This situation was expected when we refer to the forecast results in Section 3.4.1. It is also shown there that the deletion of one of the others variables such as rainfall, temperature and humidity generally reduces a bit the performance of the models in the sense of RMSE, MASE, and MARE.

### 3.7. Forecasts Results Using Saturation

In this section, we made a main modification to the rainfall variable. We call this method by saturation and the procedure is entirely detailed in Section 2.3.3 and the algorithm therein (Algorithm 2). Statistical results of this method are presented in Table 6. They permit us to distinguish the performance given by this novelty compared to Table 2. The confidence interval of the predictions in Figure 20 and Figure 21 (reproduce with Saturated_and_non_saturated_rainfall_2022_07_23.py) are not shown based on the same observations made in the previous Section 3.4.1.

As a result, an improvement of the forecasts is observed based on Table 6. For Dakar, we observed an improvement of the results after applying the saturation according to all the metrics. It is interesting to see that, in Dakar, the introduction of the saturated rainfall has reduced the over-estimation occurring at the end of 2015; compare Figure 17 and  Figure 20. Then, in Fatick, all the values are 1, meaning that the saturation method does not improve the results. That can be explained by the fact that the rainfall is less used in this region, according to Figure 18B. As for Kedougou, we have a ratio of MARE smaller than its value before applying the method (0.87) and a ratio of $R_{\mathrm{COR}}^{2}$ higher than the value before applying the method (1.04). There is no improvement in terms of ration of the RMSE and MASE. That may be explained by the fact that there was not a peak as that is illustrated in Figure 19 or the forecasts have been already good. Another favorable effect of the saturation is that, in Kedougou, the non-saturated rainfall is negatively used (Figure 19B) while the saturated rainfall is positively used (Figure 21B).

## 4. Conclusions

For three endemic regions of Senegal, we have investigated the accuracy, in the sense of the metrics such as RMSE, MASE and MARE, of the falciparum malaria incidence count per month forecasts obtained with GLM’s by using meteorological data and history of falciparum malaria incidence count per month as explanatory variables. Using the Vuong test, we have compared the adequacy of Poisson-based and NB-based GLM’s. And the Poisson with identity as a GLM link function is in practice the more adequate regression model to make forecasts of malaria incidence based on meteorological factors. We have observed that the choice of the GLM’s link function and the use of adequate lags in the explanatory variables may have a considerable impact on the forecast accuracy. We also have observed that the application of saturation in the rainfall increases the quality of the forecasts in Dakar and Kedougou.

Ablation study shows that removing the history of malaria cases from the explanatory variables has a strong adverse effect on the forecast accuracy.

This study is led with a monthly malaria incidence count due to the unavailability of the daily malaria incidence reports that could have helped us to understand better the influence of the climatic data. This study is a step towards providing the authorities with decision-making tools for the optimal dispatch of resources.

This proposed GLM gives some overestimations in the forecasts (end of 2016 in Dakar in Figure 17 and Fatick in Figure 18). These peaks can be caused by an inadequacy of this model due to its linear character even if a non-linearity is then applied for rainfall. This may also be due to a variable ratio of infected people being sufficiently ill to go to the hospital and be confirmed, thus causing a bias in the collected data. Another and important explanation to the over-forecasting (or under-forecasting) observed can be the unavailability of some explanatory variables: the distance to water bodies, the normalized differenced vegetation index (NDVI), the night and day LST (land surface temperature), the ownership and use of insecticide treated nets (ITNs) and the intermittent preventive treatment distributed for pregnant women (IPTp) that are considered to fit malaria incidence [13]. Providing the enhanced vegetation index (EVI), and actual evapotranspiration (ETa) could help to improve the malaria incidence fitting model [27]. In our available data, there was the ITNs distributed because it constitutes the main factor fighting against malaria according to [16] but its distribution was not very regular (see Figure 5, Figure 6 and Figure 7). Needless to say, the ownership and use of ITNs would be a more suitable explanatory variable.

## Figures and Tables

**Figure 1 ijerph-20-06303-f001:**
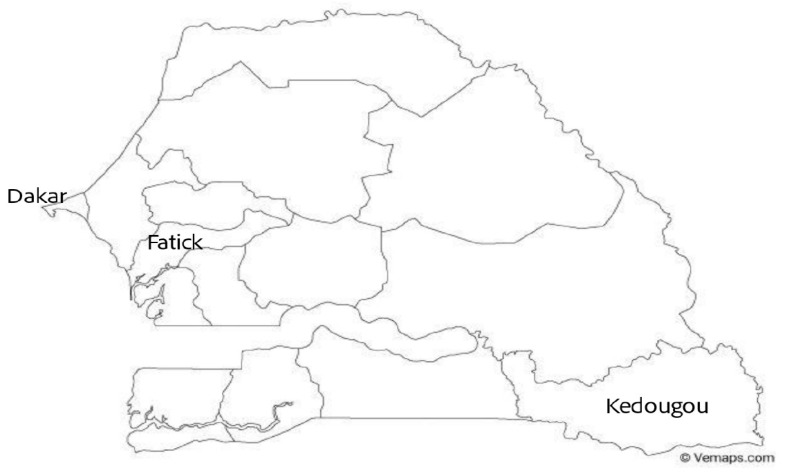
Location of Dakar, Fatick, and Kedougou in Senegal. The choice of these three regions is motivated by data availability (notably the presence of villages where longitudinal studies have been conducted), but also by geographical differences: influence of the ocean in the Dakar peninsula, tropical climate in Kedougou, and savanna landscape in Fatick. These fundamental geographical differences allow us to test the applicability of GLMs under drastically different evolutions of the climate variables.

**Figure 2 ijerph-20-06303-f002:**
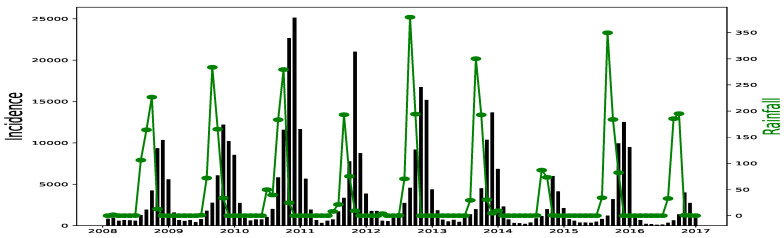
Malaria (falciparum malaria incidence count per month, black) and rainfall (mm per month, green) in Dakar, 2008–2016.

**Figure 3 ijerph-20-06303-f003:**
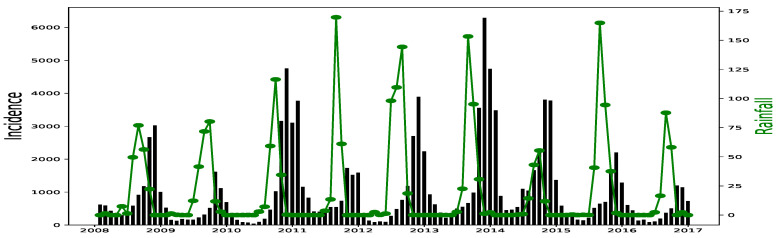
Malaria (falciparum malaria incidence count per month, black) and rainfall (mm per month, green) in Fatick, 2008–2016.

**Figure 4 ijerph-20-06303-f004:**
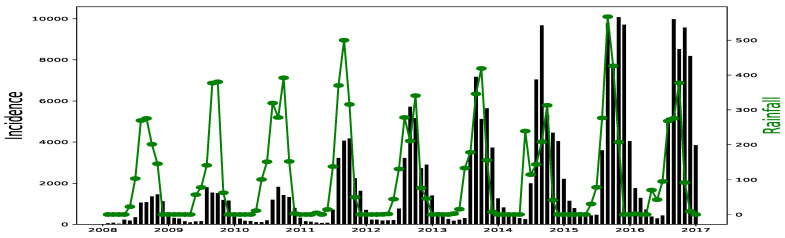
Malaria (falciparum malaria incidence count per month, black) and rainfall (mm per month, green) in Kedougou, 2008–2016.

**Figure 5 ijerph-20-06303-f005:**
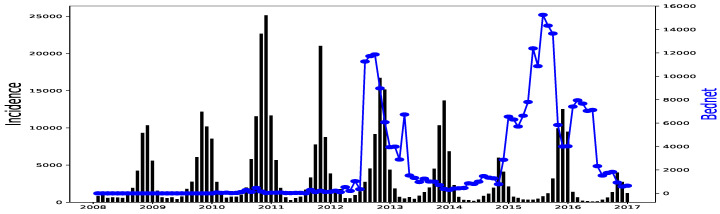
Malaria (falciparum malaria incidence count per month, black) and Bed-net distributed (the number of insecticide treated bed-nets distributed per month, blue) in Dakar, 2008–2016.

**Figure 6 ijerph-20-06303-f006:**
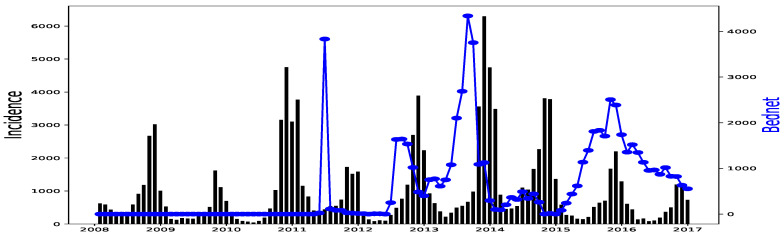
Malaria (falciparum malaria incidence count per month, black) and Bed-net distributed (the number of insecticide treated bed-nets distributed per month, blue) in Fatick, 2008–2016.

**Figure 7 ijerph-20-06303-f007:**
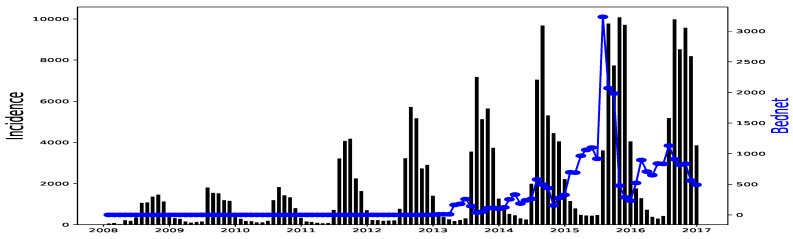
Malaria (falciparum malaria incidence count per month, black) and Bed-net distributed (the number of insecticide treated bed-nets distributed per month, blue) in Kedougou, 2008–2016.

**Figure 8 ijerph-20-06303-f008:**
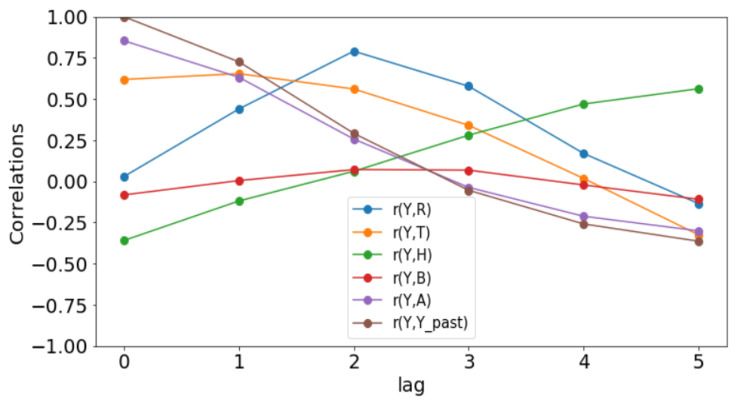
Correlation plots for Dakar.

**Figure 9 ijerph-20-06303-f009:**
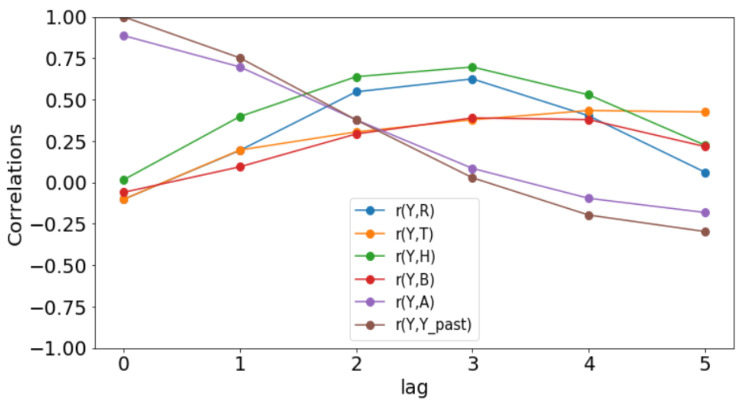
Correlation plots for Fatick.

**Figure 10 ijerph-20-06303-f010:**
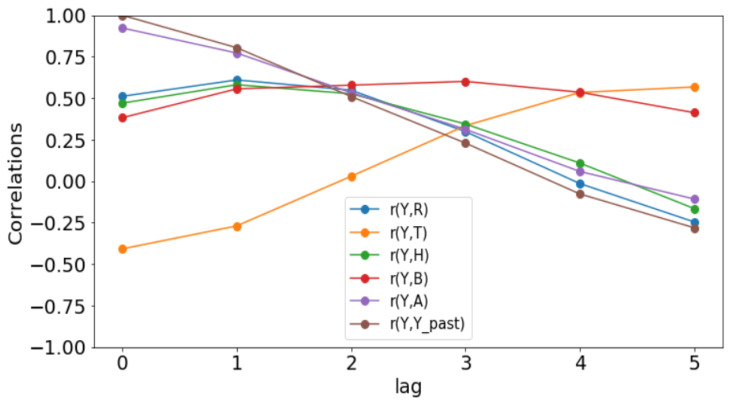
Correlation plots for Kedougou.

**Figure 11 ijerph-20-06303-f011:**
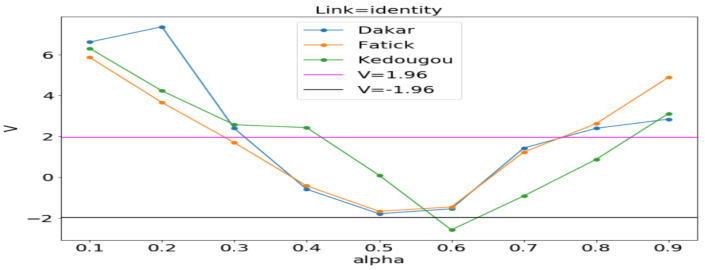
*V* vs. α between Poisson and NB distributions. Estimating α in each region by the ordinary least squares (OLS) method gives 0.109, 0.222 and 0.282, respectively, in Dakar, Fatick, and Kedougou.

**Figure 12 ijerph-20-06303-f012:**
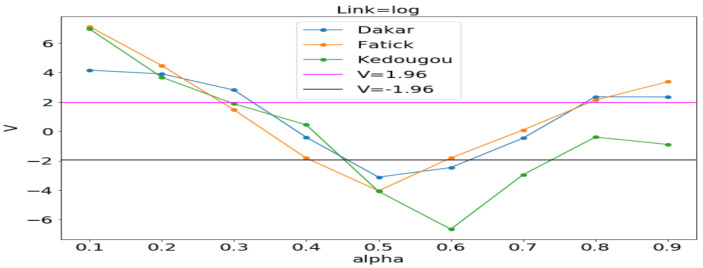
*V* vs. α between Poisson and NB distributions. Estimating α in each region by the ordinary least squares (OLS) method gives 0.142, 0.202 and 0.412, respectively, in Dakar, Fatick, and Kedougou.

**Figure 13 ijerph-20-06303-f013:**
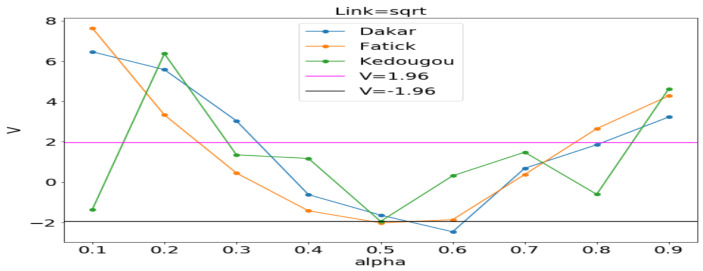
*V* vs. α between Poisson and NB distributions. Estimating α in each region by the ordinary least squares (OLS) method gives 0.119, 0.214 and 0.318, respectively, in Dakar, Fatick, and Kedougou.

**Figure 14 ijerph-20-06303-f014:**
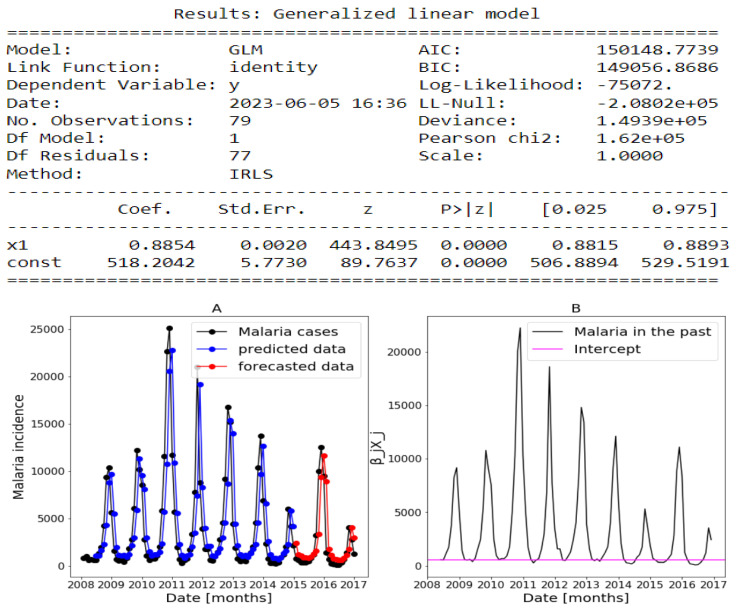
Statistical (in **top**) and forecasting (in **bottom**) results in Dakar. Malaria incidence means the falciparum malaria incidence count per month. The train/test accuracy measures are RMSE: 3886.43 /2367.55, MASE: 0.95/1.06, MARE: 0.85/1.59, and $R_{\mathrm{COR}}^{2}$: 0.51/0.54. We present the forecast results (noted by **A**) and the curves of $\beta_{j}X_{j}$ (noted by **B**).

**Figure 15 ijerph-20-06303-f015:**
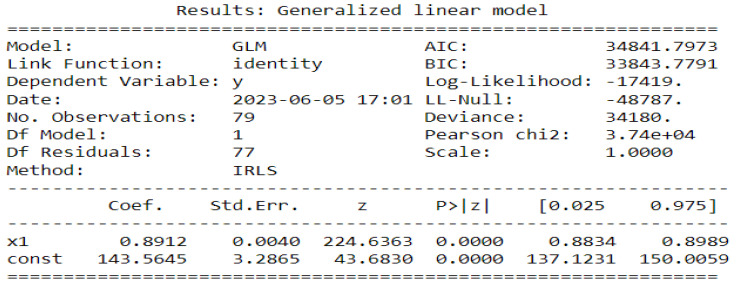
Statistical (in **top**) and forecasting (in **bottom**) results in Fatick. Malaria incidence means the falciparum malaria incidence count per month. The train/test accuracy measures are RMSE: 916.35/408.29, MASE: 0.96/1.11, MARE: 0.76/0.75, and $R_{\mathrm{COR}}^{2}$: 0.55/0.5. We present the forecast results (noted by **A**) and the curves of $\beta_{j}X_{j}$ (noted by **B**).

**Figure 16 ijerph-20-06303-f016:**
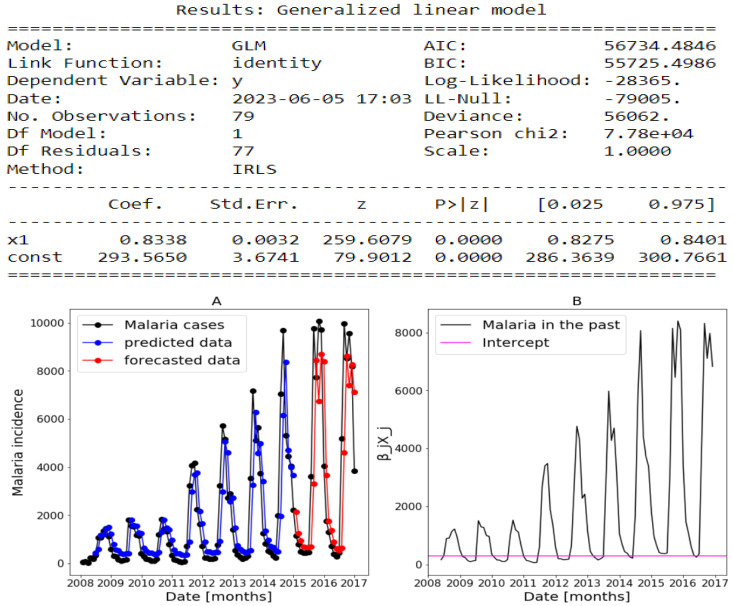
Statistical (in **top**) and forecasting (in **bottom**) results in Kedougou. Malaria incidence means the falciparum malaria incidence count per month. The train/test accuracy measures are RMSE: 1250.53/2523.74, MASE: 1.02/0.93, MARE: 1.06/0.61, and $R_{\mathrm{COR}}^{2}$: 0.61/0.59. We present the forecast results (noted by **A**) and the curves of $\beta_{j}X_{j}$ (noted by **B**).

**Figure 17 ijerph-20-06303-f017:**
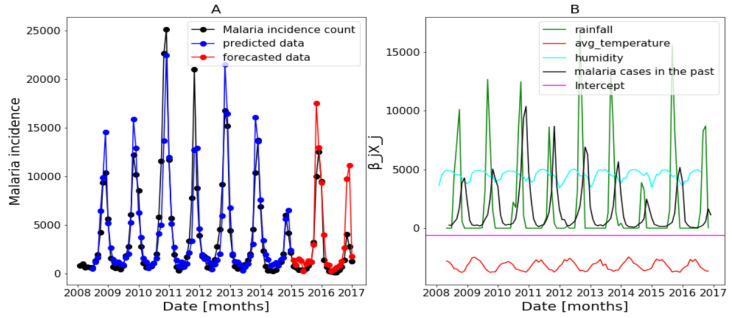
Forecasting results in Dakar, no saturation applied. Malaria incidence means the falciparum malaria incidence count per month. We present the forecast results (noted by **A**) and the curves of $\beta_{j}X_{j}$ (noted by **B**).

**Figure 18 ijerph-20-06303-f018:**
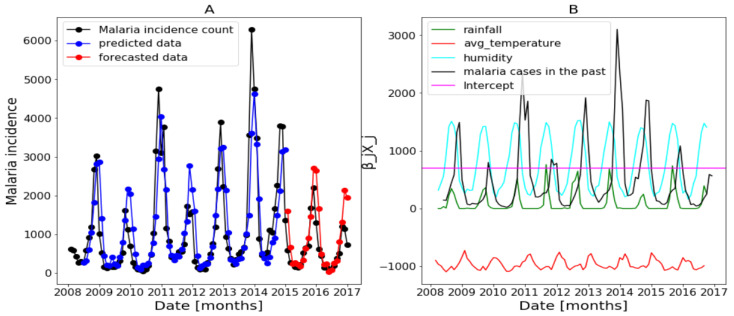
Forecasting results in Fatick: no saturation applied. Malaria incidence means the falciparum malaria incidence count per month. We present the forecast results (noted by **A**) and the curves of $\beta_{j}X_{j}$ (noted by **B**).

**Figure 19 ijerph-20-06303-f019:**
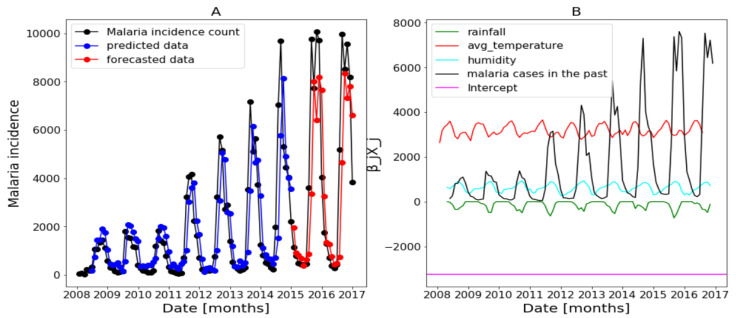
Forecasting results in Kedougou: no saturation applied. Malaria incidence means the falciparum malaria incidence count per month. We present the forecast results (noted by **A**) and the curves of $\beta_{j}X_{j}$ (noted by **B**).

**Figure 20 ijerph-20-06303-f020:**
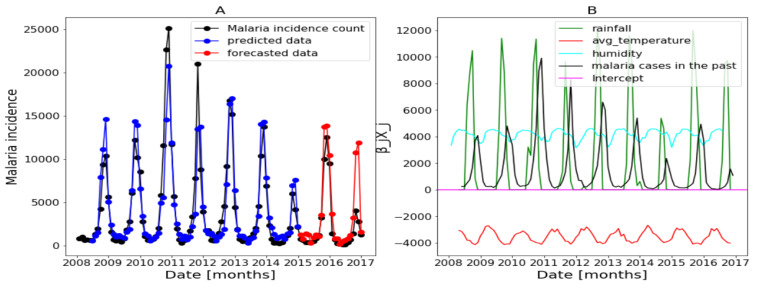
Forecasting results of the saturation in Dakar. Malaria incidence means the falciparum malaria incidence count per month. We present the forecast results (noted by **A**) and the curves of $\beta_{j}X_{j}$ (noted by **B**).

**Figure 21 ijerph-20-06303-f021:**
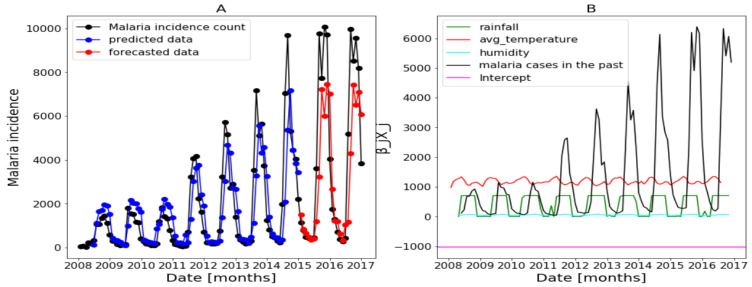
Forecasting results of the saturation in Kedougou. Malaria incidence means the falciparum malaria incidence count per month. We present the forecast results (noted by **A**) and the curves of $\beta_{j}X_{j}$ (noted by **B**).

**Table 1 ijerph-20-06303-t001:** Sample Correlations between the falciparum malaria incidence count per month and the explanatory variables in Dakar, Fatick, and Kedougou, from 2008 to 2016. These correlations are the maximum values in absolute value obtained at index *ℓ*. The statistical significance of these correlations is tested by calculating the *p*-value associated with the Pearson correlation coefficient by using the Scipy pearsonr() function, which returns the Pearson correlation coefficient along with the two-tailed *p*-value. Correlation, lag and *p*-values are reported.

	Dakar	Fatick	Kedougou
$Y_{o}\left\{t\right\}\;\mathrm{and}\;R_{t-𝓁}$	0.79	0.62	0.61
	𝓁=2	𝓁=3	𝓁=1
	$4.48\times 10^{-23}$	$2.14\times 10^{-12}$	$9.93\times 10^{-12}$
$Y_{o}\left\{t\right\}\;\mathrm{and}\;T_{t-𝓁}$	0.65	0.43	0.56
	𝓁=1	𝓁=4	𝓁=5
	$8.73\times 10^{-14}$	$5.04\times 10^{-06}$	$4.86\times 10^{-10}$
$Y_{o}\left\{t\right\}\;\mathrm{and}\;H_{t-𝓁}$	0.56	0.69	0.58
	𝓁=5	𝓁=3	𝓁=1
	$7.43\times 10^{-10}$	$5.88\times 10^{-16}$	$1.50\times 10^{-10}$
$Y_{o}\left\{t\right\}\;\mathrm{and}\;B_{t-𝓁}$	0.15	0.38	0.60
	𝓁=6	𝓁=3	𝓁=3
	0.129	$5.99\times 10^{-05}$	$3.16\times 10^{-11}$
$Y_{o}\left\{t\right\}\;\mathrm{and}\;A_{t-𝓁}$	0.85	0.88	0.92
	𝓁=0	𝓁=0	𝓁=0
	$8.61\times 10^{-30}$	$5.84\times 10^{-35}$	$6.48\times 10^{-43}$
$Y_{o}\left\{t\right\}\;\mathrm{and}\;Y_{o}\left\{t-𝓁\right\}$	0.72	0.75	0.80
	𝓁=1	𝓁=1	𝓁=1
	$9.55\times 10^{-18}$	$1.31\times 10^{-19}$	$4.28\times 10^{-24}$

**Table 2 ijerph-20-06303-t002:** Results of accuracy measures with Algorithm 1 and the explanatory variables $Y_{o}\left(t-h\right)$, $R(t-{𝓁}_{R})$, $T(t-{𝓁}_{T})$, $H(t-{𝓁}_{H})$, and 1 for the intercept. We have considered $t_{s}=0$, $t_{i}=5$, $t_{c}=84$ and $t_{e}=108$ and h=1. Refer to Table 1 for ${𝓁}_{R}$, ${𝓁}_{T}$ and ${𝓁}_{H}$ values. Train/test values are reported and $\min_{t\in \{t_{i},\ldots ,t_{c}\}}\mu \left(t\right)$. We denote by G: Gaussian, P: Poisson, id: identity, Dk: Dakar, Ft: Fatick, and Kd: Kedougou.

	Model	Link	RMSE	MASE	MARE	$R_{\mathrm{COR}}^{2}$	min	RA
Dk	G	id	2197.29/2384.26	0.52/1.01	0.68/1.54	0.84/0.79	−517.03	28.75/20
		log	2466.29/2352.81	0.6/1.38	0.85/2.66	0.79/0.78	511.81	22.5/4
	P	id	2245.27/2689.74	0.52/1.02	0.54/1.28	0.83/0.78	282.6	32.5/16
		log	2523.01/2297.45	0.57/1.23	0.65/2.05	0.79/0.77	341.22	27.5/4
		sqrt	2354.97/2303.14	0.54/1.08	0.62/1.77	0.81/0.8	279.49	30/12
	NB	id	2558.87/3555.94	0.58/1.28	0.5/1.24	0.82/0.76	460.63	27.5/16
		log	3736.91/2424.02	0.74/1.1	0.61/1.89	0.73/0.79	471.42	28.75/8
		sqrt	3409.99/3234.53	0.73/1.23	0.55/1.56	0.79/0.79	482.42	28.75/16
Ft	G	id	768.52/578.88	0.83/1.33	0.66/0.59	0.67/0.75	87.78	27.5/24
		log	721.66/484.07	0.81/1.31	0.73/0.7	0.71/0.83	83.29	25/16
	P	id	772.32/543.56	0.82/1.26	0.65/0.6	0.67/0.72	99.12	31.25/24
		log	741.92/491.42	0.83/1.37	0.85/0.88	0.69/0.78	215.44	25/8
		sqrt	763.78/526.94	0.84/1.32	0.79/0.74	0.68/0.73	215.13	26.25/20
	NB	id	839.53/527.32	0.87/1.22	0.61/0.59	0.64/0.67	69.22	30/20
		log	861.98/466.03	0.91/1.21	0.84/0.79	0.63/0.7	268.19	21.25/20
		sqrt	942.27/504.91	0.98/1.13	0.75/0.59	0.62/0.64	180.39	22.25/28
Kd	G	id	1230.61/2467.27	1.01/0.92	1.19/0.63	0.62/0.62	29.85	10/16
		log	1409.28/2720.94	1.27/1.01	2.18/0.73	0.51/0.56	872.92	13.75/20
	P	id	1241.89/2431.91	0.99/0.87	0.9/0.47	0.61/0.63	126.76	16.25/24
		log	1488.67/3009.49	1.15/1.1	1.47/0.52	0.46/0.59	510.81	21.25/16
		sqrt	1352.9/2523.28	1.03/0.89	1.07/0.42	0.56/0.62	324.21	18.75/12
	NB	id	1287.33/2346.07	1.07/0.83	0.9/0.44	0.61/0.65	−52.81	16.25/28
		log	2160.53/7024.86	1.42/2.47	1.08/0.73	0.4/0.58	275.72	16.25/8
		sqrt	1567.23/3037.79	1.19/1.14	0.91/0.44	0.54/0.63	132.16	20/16

**Table 3 ijerph-20-06303-t003:** Accuracy measures in the three regions: Algorithm 1 with Poisson for *f* and identity for *g* where the set of explanatory variables is Equartion (Equation 16) and h=1,2,3. Train/test values are reported.

Regions	*h*	RMSE	MASE	MARE	$R_{\mathrm{COR}}^{2}$	SI
Dakar	1	3886.43/2367.55	0.95/1.06	0.85/1.59	0.51/0.54	33.93/49.35
Dakar	2	5265.41/3565.7	1.47/2.15	2.4/5.97	0.08/0.06	41.5/69.91
Dakar	3	5399.89/4075.83	1.55/2.79	3.15/9.38	0.01/0.01	60.31/56.57
Fatick	1	916.35/408.29	0.96/1.11	0.76/0.75	0.55/0.5	22.22/18.83
Fatick	2	1250.88/741.17	1.47/2.26	1.99/2.21	0.14/0.03	24.92/24.75
Fatick	3	1339.06/807.71	1.67/2.71	2.85/3.32	0.0/0.01	25.23/25.19
Kedougou	1	1250.53/2523.74	1.02/0.93	1.06/0.61	0.61/0.59	30.69/36.47
Kedougou	2	1790.88/3770.72	1.58/1.59	2.67/1.23	0.19/0.18	41.54/41.69
Kedougou	3	1972.37/4427.78	1.87/1.84	3.65/1.43	0.02/0.01	42.41/47.11

**Table 4 ijerph-20-06303-t004:** Results of the addition study in the three regions: Algorithm 1 with Poisson for *f* and identity for *g*, $t_{s}=0$, $t_{i}=5$, $t_{c}=84$, $t_{e}=108$ and h=1. We denote by w: situation in the test period with the added variable and wo: situation in the test period without the indicated variable. Ratios of w/wo are reported. Note that, for the metrics such as RMSE, MASE and MARE, a ratio lower than 1 implies an improvement of the model when the variable is added as an explanatory variable. A ratio of $R_{\mathrm{COR}}^{2}$ higher than 1 indicates that adding the variable improves the forecasts.

Regions	Variable	RMSE w/wo	MASE w/wo	MARE w/wo	$R_{\mathrm{COR}}^{2}$ w/wo
Dakar	Rainfall	1.18	1.02	0.99	1.43
	Temperature	0.99	1.05	1.15	1.04
	Humidity	0.94	1.04	1.06	1.11
Fatick	Rainfall	1.24	1.22	1.04	1.34
	Temperature	1	1.06	1.08	1.1
	Humidity	1.18	1.05	0.77	1.34
Kedougou	Rainfall	0.99	0.97	0.92	1.02
	Temperature	0.96	0.95	0.88	1.05
	Humidity	0.98	0.93	0.7	1.03

**Table 5 ijerph-20-06303-t005:** Results of the ablation study in the three regions: Algorithm 1 with Poisson for *f* and identity for *g*, $t_{s}=0$, $t_{i}=5$, $t_{c}=84$, $t_{e}=108$ and h=1. We denote by wo: situation in the test period without the indicated variable and w: situation in the test period with all the explanatory variables. Ratios of wo/w are reported. Note that, for the metrics such as RMSE, MASE and MARE, a ratio lower than 1 implies that the result is good without the indicated variable. A ratio of $R_{\mathrm{COR}}^{2}$ higher than 1 indicates adding the variable improves the forecasts. We denote by Mcp: Malaria cases in the past.

	Variable	RMSE wo/w	MASE wo/w	MARE wo/w	$R_{\mathrm{COR}}^{2}$ wo/w
Dakar	Mcp	1.5	1.74	2.04	0.82
	Rainfall	0.83	1.06	1.27	0.75
	Temperature	1.05	1.07	1.11	0.98
	Humidity	1.04	1.05	1.25	0.98
Fatick	Mcp	1.48	1.63	1.58	1.11
	Rainfall	0.94	0.95	1.06	0.93
	Temperature	0.96	0.98	0.96	1.01
	Humidity	0.92	1.05	1.3	0.98
Kedougou	Mcp	1.5	1.64	1.25	1.21
	Rainfall	1	0.99	0.92	1.01
	Temperature	1.02	1.02	0.92	0.97
	Humidity	1.01	1.03	1.14	0.98

**Table 6 ijerph-20-06303-t006:** Comparison results after and before the saturation: Algorithm 2 with Poisson for *f* and identity for *g*, $t_{s}=0$, $t_{i}=5$, $t_{c}=84$, $t_{e}=108$ and h=1. We denote by w: the metric with saturation and wo: the metric without saturation, in the test period. Ratios of w/wo are reported. Note that, for the metrics such as RMSE, MASE, and MARE, a ratio lower than 1 implies an improvement of the model to make good forecasts with saturation. A ratio of $R_{\mathrm{COR}}^{2}$ higher than 1 indicates that applying the saturation improves the forecasts.

	RMSE w/wo	MASE w/wo	MARE w/wo	$R_{\mathrm{COR}}^{2}$ w/wo
Dakar	0.95	0.97	0.96	1.01
Fatick	1	1	1	1
Kedougou	1.01	1.02	0.87	1.04

## Data Availability

Source codes can be found at https://www.dropbox.com/s/7vdxopgeshlxhrc/Python%20codes%20of%20malaria%20model.zip?dl=0 (accessed on 3 July 2023).

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
