# Peer review of "Generalized Linear Models to Forecast Malaria Incidence in Three Endemic Regions of Senegal"

_ijerph, 2023, doi:10.3390/ijerph20136303_

Round 1

Reviewer 1 Report

This is a well written study on malaria, an important infection, especially in Western Africa. The study appears to be well done. In particular I was positively impressed with the critical assessment (and rejection of) of the covariables bed-nets and ACT use.  

I have just a few suggestion for the authors.

1. A map with the different locations and climate zones would make the ms more attractive.

2. A (short) discussion on the measurement error in using rapid test results as malaria cases would seem appropriate. Not everybody with a fever would use (or have access to) a rapid test. Also individuals with asymptomatic parasitaemia who have a fever for unrelated reasons may test positive.

3. Would it be possible to do separate analyses (predictions of) under-5 cases as in this age group malaria presumably causes most fatalities. 

Reviewer 3 Report

The paper constructs a generalized linear model based on Poisson and negative binomial regression models for forecasting malaria incidence by considering meteorological data and history of malaria incidence as explanatory variables. The models have been applied to three endemic regions of Senegal. The paper is well written, but this paper needs some clarifications and revisions in order to meet the IJERPH requirements. List of detailed comments below:

1. Recently, there are many published papers about GLM for forecasting epidemic incidence. What is the uniqueness or contribution of your paper here (compared to those recently published papers)? Emphasize the novelty of your contribution.

2. Lines 163 – 167: The authors have described why they choose Poisson Regression Model and NB Regression Model, namely by citing some references. I suggest the authors provide more clarifications why those regression models are suitable for their data.

3. As I understood, the authors focus on Poisson and negative binomial regression models. However, in Section 3.3.2, the authors also consider a GLM with Gaussian distribution?

Reviewer 4 Report

"Malaria is a disease caused by a parasitic infection transmitted by a mosquito (female Anophele) which is very deadly for humans" - please replace "very deadly" with actual case fatality rates

Line 27 - "fitting and forecasting the malaria cases data" - For forcasting malaria case incidence?

It is not appropriate to refer to a citation number when referring to a study. Its better to refer to the first name of the author followed by et al. to mention a study "e.g., XXX et al. showed that xxx model could predict malaria case incidence in Zimbabwe"

Line 73 - "work best with time series data that exhibited periodic or seasonal characteristics and was able to predict the seasonal trend of malaria" -In which geographical region / country?, during which calendar years?, are you referring to case incidence? , falciparum or vivax malaria?

Please do not leave incomplete sentences referring to reference numbers. You can use previous authors name followed by et al. and then give citation  (e.g., Subsequently, a temperature and precipitation dependent model for the dynamic malaria transmission model was analyzed in [22].). - was analysed by XXX et al. [22]

Line 120 - are all climatic variables normally distributed to consider the mean as representative?

How is the malaria incidence defined? per month? per year?, are all malaria types considered (vivax, falciparum)? 

The variables of bednet availability and antimalarial drug distribution are poorly defined. For example is this bednet availablity fper a 1sqkm of area or bednet availability perr 1000 households? For antimalaria treatment is this the number of treatment courses per xxx population? are you considering the number prescribed or the number of treatments reported as completed (measures compliance in addition to prescription). How did you get these anti-malaria drug distribution data?

From the description alone it is difficult to appreciate the geographical areas considered. A map of senegal to scale highlighting the areas studied would be helpful

Figures 3 and 4 doesnt have units

The way the bed net (please correct the variable to bed net availability rather than "bed net" in the manuscript) availability is presented is confusing in figure 4. Nets do not disappear overtime. So is this the % of people sleeping under a bednet at any given time in the population? Again giving units would have been more helpful.

How are your training and validation datasets defined?

This manuscript needs extensive editing for language corrections before being published. The authors use gramatically correct language but the choice of words needs to improve according accepted norms of scientific writing. I assume that the manuscript was first drafted in French and then translated to English. For example phrases such as "very bad accuracy" (line 293) are innapropriate in scientific writing

Why do the data points extend to 2017 only. I suggest forcasting the malaria incidence between 2017 - 23 from the best performing models derived and then comparing it with actual reported figures as an additional validation

Round 2

Reviewer 2 Report

This version is an improved version of the latest revision, in my opinion should be published.

Author Response

We are grateful to the Reviewer for accepting this manuscript to be published.

Reviewer 3 Report

I found that the author has managed to work through all the previous comments and suggestions.  Congratulation.

Author Response

(The authors gave the same response as above.)

Reviewer 4 Report

The units must be presented in graphs either in x and y axis or in the explanation. Them being explained in text is inadequate as figures and tables are expected to have all the data and explanations for the reader so that they do not have to back and forth between the text. For example if the malaria cases are falciparum and are counted per month please label the axis as falciparum malaria incidence per month rather than as "incidence". Please also specificy that bed nets are insecticide treated bednets rather than as "bednets".
